# DYNAMIC-TREERPO: BREAKING THE INDEPENDENT TRAJECTORY BOTTLENECK WITH STRUCTURED SAMPLING

## ABSTRACT

The integration of Reinforcement Learning (RL) into flow matching models for text-to-image (T2I) generation has driven substantial advances in generation quality. However, these gains often come at the cost of exhaustive exploration and inefficient sampling strategies due to slight variation in the sampling group. Building on this insight, we propose Dynamic-TreeRPO, which implements the sliding-window sampling strategy as a tree-structured search with dynamic noise intensities along depth. We perform GRPO-guided optimization and constrained Stochastic Differential Equation (SDE) sampling within this tree structure. By sharing prefix paths of the tree, our design effectively amortizes the computational overhead of trajectory search. With well-designed noise intensities for each tree layer, Dynamic-TreeRPO can enhance the variation of exploration without any extra computational cost. Furthermore, we seamlessly integrate Supervised Fine-Tuning (SFT) and RL paradigm within Dynamic-TreeRPO to construct our proposed LayerTuning-RL, reformulating the loss function of SFT as a dynamically weighted Progress Reward Model (PRM) rather than a separate pretraining method. By associating this weighted PRM with dynamic-adaptive clipping bounds, the disruption of exploration process in Dynamic-TreeRPO is avoided. Benefiting from the tree-structured sampling and the LayerTuning-RL paradigm, our model dynamically explores a diverse search space along effective directions. Compared to existing baselines, our approach demonstrates significant superiority in terms of semantic consistency, visual fidelity, and human preference alignment on established benchmarks, including HPS-v2.1, PickScore, and ImageReward. In particular, our model outperforms SoTA by $4.9\%$, $5.91\%$, and $8.66\%$ on those benchmarks, respectively, while improving the training efficiency by nearly $50\%$.

## 1 INTRODUCTION

Flow matching-based image generation models (Lipman et al., 2022; Liu et al., 2022; Esser et al., 2024; Labs, 2024; Batifol et al., 2025), renowned for their solid theoretical foundations and impressive performance, have demonstrated remarkable results in text-to-image tasks. However, significant challenges remain in scenarios involving text rendering, numerals, and fine-grained attribute control. Recent advances (Ouyang et al., 2022; Fan et al., 2023; Xu et al., 2023; Wallace et al., 2024; Gong et al., 2025) have shown that incorporating GRPO (Shao et al., 2024) in the pretraining phase can lead to improved performance. However, the inherent trial-and-error nature of reinforcement learning fundamentally limits the efficiency and effectiveness of these approaches.

Current GRPO-based probability flow models (Liu et al., 2025; Xue et al., 2025; Li et al., 2025; He et al., 2025) introduce stochasticity at each time step via stochastic differential equations, and leverage GRPO to optimize the entire state-action sequence. However, these approaches incur substantial computational overhead during the exploratory denoising process, significantly slowing down the training speed. Subsequent methods such as MixGRPO (Li et al., 2025) and Tempflow-GRPO (He et al., 2025) accelerate training by reducing the number of SDE sampling in denoising process, leading to slight variation and similar trajectories in sampling group. Meanwhile, the prevailing approaches (Bai et al., 2025; Chu et al., 2025; Zhang et al., 2025b) still adhere to the sequential

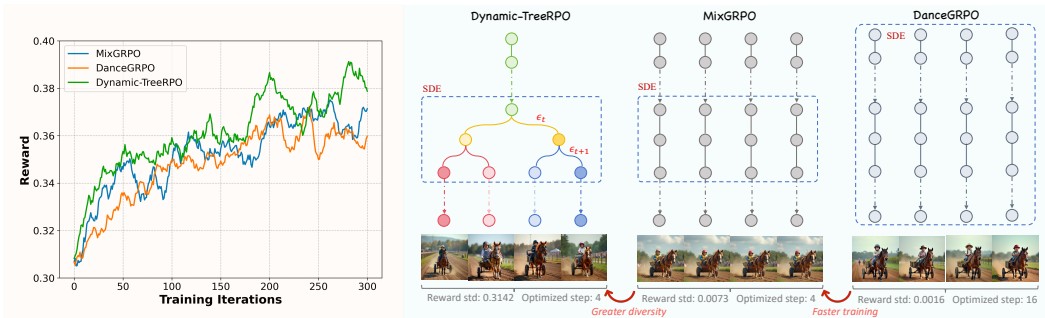

Figure 1: Compare with the previous method. $Left$: The reward curve during training shows that Dynamic-TreeRPO converges more rapidly than both DanceGRPO and MixGRPO, and ultimately achieves significantly better results than either of them. $Right$: Visualization of the different structures. Dynamic-TreeRPO employs a tree structure with a sliding window mechanism. MixGRPO utilizes a sliding window structure, where SDE is applied only during the sliding window period. In contrast, DanceGRPO applies SDE throughout the entire process.

SFT-then-RL paradigm. This fully decoupled two-stage setup is susceptible to catastrophic forgetting, inefficient exploration, and hallucinations (Lv et al., 2025).

To address these challenges, we propose Dynamic-TreeRPO, an intuitive and effective solution that integrates a hybrid ODE-SDE strategy into the framework composed of a sliding window and a tree structure. Our approach enables diversified sample generation with shared prefixes of sampling trajectories. Specifically, we implement the SDE sampling strategy within the sliding window as tree structure and employ ODE sampling elsewhere, thereby confining the stochasticity in this tree. The common inference steps of the ancestor nodes in the tree are computed once and shared for all descendants, pursing for efficient computation. To expand the exploration of tree-structured sampling, we design a dynamic noise intensity along the depth of the tree. As observed in Figure 1, the reward variance of Dynamic-TreeRPO is larger than that of other methods, indicating a broader exploration space of our method.

Furthermore, we introduce a novel training strategy, denoted LayerTuning-RL, to seamlessly integrate SFT with Dynamic-TreeRPO. SFT primarily learns by exploiting high-quality expert data, while RL explores through interaction and feedback from the environment. The conventional approach combines these two by first performing SFT and then RL. Although intuitive, this often yields suboptimal results in practice, as the RL phase consistently suffers from issues such as policy hacking and catastrophic forgetting (Chen et al., 2025b; Zhang et al., 2025a). The underlying reason is the disconnection between SFT and RL in the objective function, representation space, and exploration mechanism. Generally, the RL stage lacks the capability to continuously anchor to the knowledge acquired in SFT.

To address this, we rethink SFT not as an independent training method, but as a dynamically weighted auxiliary objective within the RL process. We apply the modified SFT function at each layer of the tree structure, armed with which the Dynamic-TreeRPO can perform more efficiently and robustly for the exploration of diverse sampling. In this way, the SFT function acts as a PRM to guide the sampling direction of each decision node. Additionally, we decouple the clipping bounds in the GRPO algorithm and dynamically adjust them based on the training steps, which prevents entropy collapse and fully leverages the potential of GRPO. In summary, our contributions are as follows.

- We propose Dynamic-TreeRPO, a novel RL training framework that formulates the sampling process as a tree-structured search. By employing the sliding window strategy and a tree-structured search with dynamic noise mechanism, our method amortizes the computational cost across shared tree prefixes for the purpose of efficient training.

- We introduce a new training paradigm, LayerTuning-RL, which seamlessly integrates SFT and the proposed Dynamic-TreeRPO. Rather than treating SFT as an isolated training phase, we reformulate it as a dynamically weighted auxiliary objective throughout the

training process, effectively mitigating issues such as catastrophic forgetting, inefficient exploration, and model hallucinations.

- Our method improves the training efficiency of T2I task by nearly $50\%$ over the prior state-of-the-art and achieves superior performance on several benchmarks in terms of semantic consistency, visual quality, and human preference alignment.

## 2 RELATED WORK

Diffusion models (Ho et al., 2020; Song et al., 2020b;a; Lu et al., 2025; 2022; Zheng et al., 2023; Zhao et al., 2023; Salimans & Ho, 2022) gradually add noise to data until it becomes random noise, then learn to reverse this process. Sampling is performed using either discrete DDPM steps or probabilistic flow SDE solvers to generate high-fidelity outputs. Flow matching (Lipman et al., 2022; Yin et al., 2024; Gao et al., 2024) constructs a continuous path between the noise and data distributions by directly matching the velocity fields, enabling the learning of a continuous time-normalized flow, so that only a few ODE steps are sufficient for deterministic sampling. Recent works, e.g., Flow-GRPO (Liu et al., 2025) and Dance-GRPO (Xue et al., 2025), introduce GRPO into flow matching models, converting deterministic flow models into equivalent SDEs via ODE-to-SDE conversion strategies, while preserving the marginal distributions of the original models to support RL-based stochastic sampling. However, previous methods, e.g., Flow-GRPO (Liu et al., 2025), still suffer from inefficiency issues, as they require sampling and optimization over all denoising steps. MixGRPO (Li et al., 2025) addresses this problem by proposing a mixed sampling strategy and introducing a sliding window mechanism, where SDE sampling and GRPO-guided optimization are only applied within the window. TempFlow-GRPO (Xue et al., 2025) introduces a trajectory branching mechanism that provides process rewards at designated branching points to enable precise credit assignment without requiring dedicated intermediate reward models. Nevertheless, these methods still face challenges such as insufficient intra-group diversity and high computational overhead for group operations. To address these issues, we propose an elegant and intuitive solution, Dynamic-TreeRPO, which reconstructs the processes of different samples into a binary tree with shared prefixes and a sliding window mechanism, based on a hybrid ODE-SDE strategy.

Previous methods (Chu et al., 2025; Chen et al., 2025a) have explored the differences between SFT and RL. (Chu et al., 2025) observed that in both visual and rule-based textual environments, RL trained with outcome-based rewards achieves better generalization, while SFT tends to specialize to the specific data distribution, exhibiting memorization of the training data. Consequently, some recent works have focused on hybrid training paradigms to harness their complementary benefits. In the field of LLMs, a two-stage approach (Chen et al., 2025b) has been introduced into the training process, which leverages the synergy between RL and SFT to enhance model performance. For T2I task, SimpleAR (Wang et al., 2025) applies SFT to enhance fidelity and instruction-following capability, and then uses RL to further refine multimodal alignment and mitigate bias. However, this completely decoupled two-stage approach (SFT followed by RL) tends to forget the knowledge acquired during the SFT phase and can lead to inefficiencies in the exploration space of RL. To address this, we are the first to propose a novel training strategy, LayerTuning-RL, for flow-matching-based image generation tasks, which integrates supervised fine-tuning with Dynamic-TreeRPO.

## 3 METHOD

In this paper, we aim to establish an efficient, stable, and robust RL paradigm for T2I generation. By equipping the sliding window strategy with tree structure sampling, we enable prefix sharing among tree branches to amortize computational overhead. Meanwhile, we propose a novel training paradigm, *i.e.* LayerTuning-RL, to integrate SFT function as PRM for the stable training of RL paradigm. In Section 3.1, we first introduce the main idea of GRPO on flow matching model. Then, we study the weaknesses of GRPO, such as inefficiency and vulnerability, to reward hacking, and propose the Dynamic-TreeRPO in Section 3.2. In Section 3.3, we describe the dynamic optimization of the upper and lower clipping ranges for GRPO. Finally, we present the framework of our LayerTuning-RL in Section 3.4.

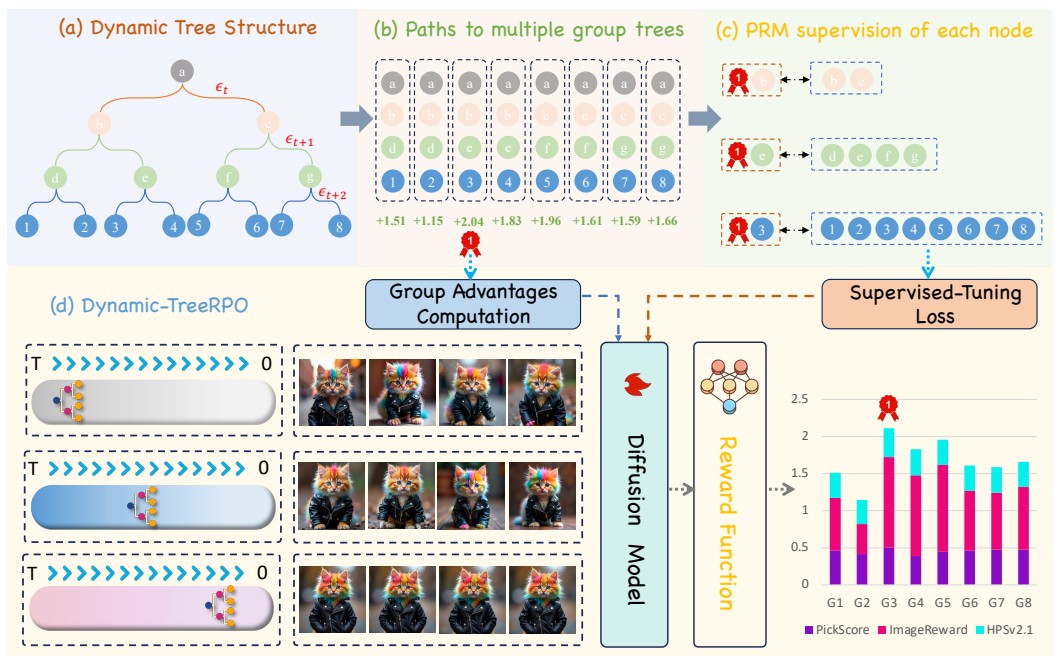

Figure 2: The framework of Dynamic-TreeRPO. ($a$) Dynamic Tree Structure. Noise intensity is dynamically introduced for the nodes of each layer in the tree structure. ($b$) Paths to multiple group trees. For each path, the highest reward score is selected. ($c$) PRM supervision of each node. The node with the maximum reward is used to supervise the model's predictions at each layer. ($d$) Training procedure of Dynamic-TreeRPO.

## 3.1 PRELIMINARY: FLOW-BASED GRPO

In this section, we introduce the core concept of GRPO, and then review how flow-based GRPO converts a deterministic ODE sampler into an SDE sampler with the same marginal distribution to meet the stochastic exploration requirements of GRPO.

**GRPO on Flow Matching**. RL aims to learn a policy that maximizes the expected cumulative reward by optimizing the policy model to maximize the following objective:

$$\mathcal{J}(\theta) = \mathbb{E}_{c \sim C, \{\mathbf{x}_i\}_{i=1}^G \sim \pi_{\theta_{old}}(\cdot|c)} \left[ \frac{1}{G} \sum_{i=1}^G \frac{1}{T} \sum_{t=1}^T \min \left( \rho_{t,i} A_i, \text{clip}(\rho_{t,i}, 1-\varepsilon, 1+\varepsilon) A_i \right) \right], \quad (1)$$

where $\rho_{t,i} = \frac{\pi_\theta(\mathbf{x}_{t-1,i}|\mathbf{x}_{t,i},c)}{\pi_{\theta_{old}}(\mathbf{x}_{t-1,i}|\mathbf{x}_{t,i},c)}$, $T$ is the timestep, and $\pi_\theta(\mathbf{x}_{t,i-1}|\mathbf{x}_{t,i},c)$ is the policy function used to generate the output at time step $t$ in a Markov Decision Process (MDP). $\varepsilon$ is a hyperparameter. $A_i$ denotes the advantage function. Given a prompt $\mathbf{c}$, the model samples a set of $G$ images $\{\mathbf{x}_{0,i}\}_{i=1}^G$, and obtains the corresponding rewards $R = \{r_1, r_2, \ldots, r_G\}$, which are then standardized as:

$$A_i = \frac{r_i - \text{mean}(\{r_1, r_2, \cdots, r_G\})}{\text{std}(\{r_1, r_2, \cdots, r_G\})}. \quad (2)$$

**Convert ODE to SDE.** GRPO requires stochastic exploration through multiple trajectory samples, where policy updates depend on the probability distribution of trajectories and their associated reward signals. However, flow matching models utilize deterministic ODE sampling:

$$\mathrm{d}\mathbf{x}_t = \mathbf{v}_t \mathrm{d}t, \quad (3)$$

where $v_t$ denotes the velocity field. Flow-GRPO and DanceGRPO convert the sampling process of rectified flows from a deterministic ordinary differential equation (ODE) to an equivalent stochastic differential equation (SDE), ensuring that the marginal probability density function at all time steps remains consistent with that of the original model. The SDE sampling process can be formulated as

follows:

$$d\mathbf{x}_t = \left(\mathbf{v}_t - \frac{1}{2}g_t^2 \nabla \log p_t(\mathbf{x}_t)\right) dt + g_t d\mathbf{w}, \tag{4}$$

where $d\mathbf{w}$ denotes Brownian motion, $\nabla \log p_t(\mathbf{x}_t)$ represents the score function at time $t$, and $g(t)$ is the standard deviation of the noise. The final update rule is given by:

$$\mathbf{x}_{t+\Delta t} = \mathbf{x}_t + \left[\mathbf{v}_\theta(\mathbf{x}_t, t) + \frac{\sigma_t^2}{2t}\left(\mathbf{x}_t + (1-t)\mathbf{v}_\theta(\mathbf{x}_t, t)\right)\right]\Delta t + \sigma_t \sqrt{\Delta t}\,\epsilon, \tag{5}$$

where $\epsilon \sim \mathcal{N}(0, I)$ is used to inject stochasticity, and $\sigma_t = a\sqrt{\frac{t}{1-t}}$. Notably, previous GRPO methods typically use a KL regularization term to prevent reward over-optimization. In our experiment, we demonstrate the promising performance of LayerTuning-RL on this problem without KL regularization, which provides an alternative solution to the problem of reward over-optimization.

## 3.2 DYNAMIC-TREERPO

Typically, previous RL approaches generate multiple independent trajectories from the same initial noise to facilitate exploration. However, this practice suffers from two major drawbacks: computational redundancy and limited exploration space. Even with SDE applied at each step, the resulting trajectories often exhibit high similarity. To address this, we construct a binary tree-structured search space, where trajectories share a common prefix before branching into distinct subsequent paths. Modeling sequence generation as a binary tree search process is more than feasible and highly advantageous to efficiency. By calculating and storing the shared prefix once, Dynamic-TreeRPO can effectively avoid redundant computations for the calculation of descendant trajectories. Within tree structure, each branching node can be considered a pivot for exploration, enabling efficient and controllable expansion of inference paths. Furthermore, we introduce different noise intensities into each tree layer to enhance intra-group diversity and thus expand the searching space.

As shown in Figure 1, we follow MixGRPO to combine SDE and ODE sampling. For a tree of depth $d$, we put it in the sliding window across the denoising time range $S_{\text{tree}} = [\tau, \tau + d]$, where $\tau \in [0, T - d]$ and $T$ is the total number of denoising steps. During the denoising process, SDE sampling is employed within the tree, while ODE sampling is used outside the tree. The path from the initial time step to the root node is computed only once, which significantly reducing the forward propagation overhead. To further enhance the diversity of trajectories within each group, we introduce a differentiable noise intensity function at $k$-th tree layer:

$$g_t(k) = g_t \times (1 + \beta\frac{k}{d}), \tag{6}$$

where $\beta$ is a hyperparameter controlling the rate of noise growth, and $\frac{k}{d}$ is the normalized depth, ensuring that the noise intensity increases linearly along the depth of tree. In Dynamic-TreeRPO, the combined ODE and SDE sampling within the tree can be formulated as:

$$d\mathbf{x}_t = \begin{cases} \left(\mathbf{v}_t - \frac{1}{2}g_t^2 \nabla \log p_t(\mathbf{x}_t)\right) dt + g_t(k) d\mathbf{w}, & \text{if } t \in S_{tree} \\ \mathbf{v}_t dt, & \text{otherwise} \end{cases} \tag{7}$$

and thus the denoising update can be optimized as:

$$\mathbf{x}_{t+\Delta t} = \begin{cases} \mathbf{x}_t + \left[\mathbf{v}_\theta(\mathbf{x}_t, t) + \frac{\sigma_t^2}{2t}\left(\mathbf{x}_t + (1-t)\mathbf{v}_\theta(\mathbf{x}_t, t)\right)\right]\Delta t + \sigma_t \sqrt{\Delta t}\,\epsilon, & \text{if } t \in S_{tree} \\ \mathbf{x}_t + \mathbf{v}_\theta(\mathbf{x}_t, t)\Delta t, & \text{otherwise} \end{cases} \tag{8}$$

Finally, the training objective is written as:

$$\mathcal{J}(\theta) = \mathbb{E}_{c \sim C, \{\mathbf{x}_i\}_{i=1}^{2^{d-1}} \sim \pi_{\theta_{\text{old}}}(\cdot|c)}\left[\frac{1}{2^{d-1}}\sum_{i=1}^{2^{d-1}}\frac{1}{|S_{tree}|}\sum_{t=\tau}^{\tau+d}\right.$$
$$\left. \min\left(\rho_{t,i}A_i, \text{clip}\left(\rho_{t,i}, 1-\varepsilon, 1+\varepsilon\right)A_i\right)\right], \tag{9}$$

where the policy ratio $\rho_{t,i}$ and advantage function $A_i$ are consistent to previous settings (Li et al., 2025). Intuitively, the concept of sampling group is determined by the trajectories of binary tree.

Compared with previous methods, our optimization is performed only in the tree, rather than across all time steps. The number of function evaluations (NFE) for $\pi_{\theta_{old}}$ is reduced from $2^{d-1} \times T$ to $\tau + 2^{d-1} \times (T - \tau)$. The pseudo-code of our method can be found in Algorithm 1. In summary, we achieve significant improvements on both computational efficiency and exploration diversity through the dynamic weighted tree-structured sampling approach.

---

**Algorithm 1** DYNAMIC-TREERPO Training Algorithm

---

**Input:** Policy model $\pi_\theta$, reward models $\{R_k\}_{k=1}^K$, prompt dataset $\mathcal{C}$, total sampling steps $T$, Tree depth $d$.

**Output:** Optimized policy model $\pi_\theta$

1: **for** training iteration $m = 1$ to $M$ **do**
2:     Sample batch prompts $\mathcal{C}_b \sim \mathcal{C}$
3:     Update old policy: $\pi_{\theta_{old}} \leftarrow \pi_\theta$
4:     **for** each prompt $\mathbf{c} \in \mathcal{C}_b$ **do**
5:         Generate $2^{d-1}$ samples: $\{\mathbf{o}_i\}_{i=1}^{2^{d-1}} \sim \pi_{\theta_{old}}(\cdot|\mathbf{c})$ with tree structured sampling
6:         Compute rewards $\{r_i^k\}_{i=1}^G$ using $R_k$
7:         Find index of sample with maximum advantage: $i^* = \arg\max\{r_i^k\}_{i=1}^G$
8:         **for** each sample $i = 1$ to $2^{d-1}$ **do**
9:             **for** sampling timestep $t = 0$ **to** $T - 1$ **do**
10:               **if** $t \in Tree$ **then**
11:                   Use ree structured Sampling with to get $\mathbf{x}_{t+1}^i$ with different noise intensities
12:               **else**
13:                   Use ODE Sampling to get $\mathbf{x}_{t+1}^i$
14:               **end if**
15:               Calculate multi-reward advantage: $A_i \leftarrow \sum_{k=1}^K \frac{r_i^k - \mu^k}{\sigma^k}$
16:             **end for**
17:         **end for**
18:         **for** each timestep $t \in Tree$ **do**
19:             Update policy via gradient ascent: $\theta \leftarrow \theta + \eta \nabla_\theta \mathcal{J}(r_{i_*}^k)$
20:         **end for**
21:     **end for**
22:     Update Tree root node position
23: **end for**

---

### 3.3 DYNAMIC-ADAPTIVE CLIPPING BOUNDS FOR TRAJECTORY REWARD

Clipped probability ratio has been widely adopted in reinforcement learning to stabilize policy updates and prevent excessively large and unstable parameter changes during optimization. By introducing a hyperparameter $\epsilon$, the probability ratio is constrained within fixed clipping boundaries, as shown in the following equation:

$$|\rho_{t,i} - 1| = |\frac{\pi_\theta(\mathbf{x}_{t-1,i}|\mathbf{x}_{t,i}, c)}{\pi_{\theta_{old}}(\mathbf{x}_{t-1,i}|\mathbf{x}_{t,i}, c)} - 1| \leq \varepsilon. \tag{10}$$

For flow matching models, this constraint implies that an identical restriction is imposed on the relative change of policy outputs at every time steps. Such a "one-size-fits-all" strategy suffers from a fundamental drawback that it arbitrarily restricts the exploration with low-probability without considering its potentiality. Therefore, the capability of learning new knowledge and discovering diverse reasoning paths may be suppressed under this setting. Some methods have been proposed in the text generation domain to address this kind of issue. For example, DAPO uses asymmetric clipping boundaries, and DCPO dynamically associates the token's own probability with the clipping boundary. However, these approaches still retain some fundamental limitations or even inapplicable to the T2I task. To this end, we introduce the reward value of each trajectory into the clipping boundary, as shown in the following equation:

$$\varepsilon_t = \varepsilon_{low} + (\varepsilon_{high} - \varepsilon_{low}) \cdot e^{-\eta R_{(i)}}, \tag{11}$$

where $R_{(i)}$ is the reward value corresponding to the trajectory of the $i$-th leaf node where $i \in [0, 2^{d-1}]$, $\eta$ is the reward sensitivity factor, and $\varepsilon_{\text{low}}, \varepsilon_{\text{high}}$ are the lower and upper clipping thresholds pre-set following DAPO. Equation 11 allows the clipped probability ratio to be dynamically

adjusted within $[\varepsilon_{\text{low}}, \varepsilon_{\text{high}}]$ according to the trajectory's reward value, permitting bolder exploration steps in low-reward regions while adopting more cautious fine-tuning in high-reward regions.

### 3.4 LAYERTUNING-RL

Current training paradigms of T2I models employ RL or SFT approaches. Generally, SFT method enables the model to learn expert-level reasoning trajectories, while RL allows the model to autonomously explore and select the aligned inference paths within the existing knowledge. Some works sequentially employ SFT and RL methods, which often result in catastrophic forgetting. To better integrate the advantages of both paradigms, we propose LayerTuning-RL, a tightly coupled framework combining supervised-tuning and MIXTree-GRPO. Specifically, we introduce the following supervised-tuning objective at each time step $t$:

$$\mathcal{J}(\theta)_{\text{SFT}} = \mathbb{E}\left[\|V_t^* - V_{t,i}^\theta\|_2^2\right], \tag{12}$$

where $V_t^* = V_{t,\arg\max(R)}^\theta$ denotes the best predicted trajectory at time step $t$, which obtains the highest reward score in the set of $V_{t,i}^\theta$. In the optimization of tree nodes, $V_i^*$ is used as the pseudo target of supervised-tuning objective, which provides the guidance for the optimization directions of each layer in one sampling group. To some extent, this supervised-tuning objective can be considered as PRM for each layer in RL paradigms, and thus construct the paradigm of our LayerTuning-RL. The overall training objective of LayerTuning-RL can be written as:

$$\mathcal{J}(\theta)_{\text{fusion}} = \mathcal{J}(\theta) + \lambda \times \mathcal{J}(\theta)_{\text{SFT}}, \tag{13}$$

where $\lambda$ denotes the hyperparameter for fusing the SFT and RL paradigms.

LayerTuning-RL is implemented as a collaborative framework, where RL provides overall trajectory-level supervision based on reward optimization, and supervised-tuning offers finer-grained supervision at each node. We claim that the design of LayerTuning-RL offers three main advantages: (1) it avoids catastrophic forgetting caused by two-stage training; (2) it improves path exploration efficiency through PRM-based supervised-tuning guidance; and (3) it guarantees the performance improvement of RL by strategically transferring beneficial knowledge from supervised-tuning to LayerTuning-RL. More details can be discussed in experiments.

## 4 EXPERIMENTS

### 4.1 EXPERIMENT SETUP

**Dataset.** We evaluate Dynamic-TreeRPO on the HPDv21 dataset (Wu et al., 2023). The training set contains 103,700 prompts. However, we achieve state-of-the-art performance using only a small fraction of the training data in practice. The test set consists of 400 prompts.

**Evaluation Metrics.** We align with MixGRPO and assess performance on two metrics: computational cost and generation quality. To measure the computational cost, we report the number of function evaluations (NFE) and the average training time per GRPO iteration, which faithfully reflect the actual training overhead. For quality assessment, we employ three reward models, including HPS-v2.1, PickScore (Kirstain et al., 2023), ImageReward (Xu et al., 2023), as our evaluators.

**Implementation Details.** We use FLUX.1-dev as the base model, an advanced text-to-image diffusion model based on flow matching. The trajectory tree is configured with depth $d = 4$, yielding 8 leaf nodes. The noise growth magnitude $\beta$ is set to 0.7, and the reward sensitivity factor $\eta$ for clipping probability ratios is set to 0.5. The clipping thresholds $\varepsilon_{\text{low}}$ and $\varepsilon_{\text{high}}$ are set to $5 \times 10^{-5}$ and $5 \times 10^{-3}$, respectively. The fusion coefficient $\lambda$, which balances supervised fine-tuning and the reinforcement learning paradigm, is set to 0.02.

During training, for each prompt, we generate 8 images (one per leaf node), and each image is sampled by $T = 25$ denoising steps. The model is fine-tuned for 100 iterations on 8 NVIDIA H100 GPUs with a global batch size of 16. We use the AdamW optimizer (Loshchilov & Hutter, 2017) with a learning rate of $5 \times 10^{-6}$ and weight decay $1 \times 10^{-4}$. Training is conducted in `bfloat16` mixed precision. All other hyper-parameters are kept consistent to those of MixGRPO and other baselines.

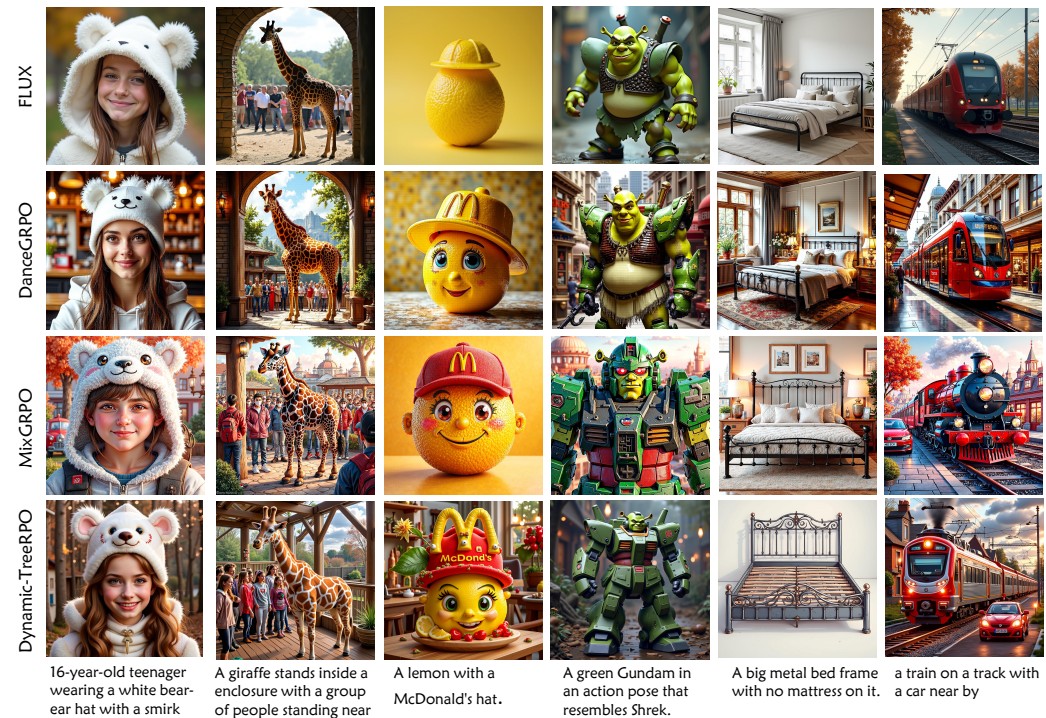

Figure 3: Qualitative comparison. Dynamic-TreeRPO achieves superior performance compared to Flux,DanceGRPO and MixGRPO in terms of semantics, aesthetics and text-image alignment.

## 4.2 MAIN EXPERIMENTS

We conduct comprehensive experiments to evaluate the training efficiency and generation quality of our Dynamic-TreeRPO. In the left part of Figure 1, the reward curve of our method consistently surpasses those of MixGRPO and DanceGRPO, indicating a faster stabilization and higher reward levels in earlier stages. The training effectiveness of our method is also evidenced in Table 1, where it achieves an average training iteration time of 151 seconds, a reduction of $114\%$ compared to DanceGRPO. More importantly, our method also achieves impressive qualitative generation results. This is evident in Figure 3, which shows that our approach exhibits a stronger ability for semantic alignment (*e.g.*, "Gundam resembles Shrek" of column 4, and not vice versa) and instruction following (e.g., "no mattress on it" of column 5). As presented in Table 1, our method significantly surpasses the four baseline methods on all three evaluation metrics.

Table 1: Comparison results of computational efficiency and image quality. Dynamic-TreeRPO achieves the best performance across multiple metrics, with the top result in each column highlighted in **bold**. In Dynamic-TreeRPO, we report the average NFE per sample.

| Method | $NFE_{\pi_{\theta_{old}}}$ | $NFE_{\pi_\theta}$ | Iteration Time (s)↓ | Human Preference Alignment | | |
| --- | --- | --- | --- | --- | --- | --- |
| | | | | HPS-v2.1↑ | Pick Score↑ | ImageReward↑ |
| FLUX | / | / | / | 0.313 | 0.227 | 1.088 |
| DanceGRPO | 25 | 14 | 323 | 0.356 | 0.233 | 1.436 |
| | 25 | 4 | 241 | 0.334 | 0.225 | 1.335 |
| MixGRPO | 25 | 4 | 240 | 0.367 | 0.237 | 1.629 |
| MixGRPO-Flash | 16 | 4 | 166 | 0.358 | 0.236 | 1.528 |
| Dynamic-TreeRPO | 13.8(Avg) | 4 | **151** | **0.385** | **0.251** | **1.770** |

### 4.3 ABLATION EXPERIMENTS

We have meticulously designed a series of experiments. Table 2 presents a comprehensive ablation study of the proposed Dynamic-TreeRPO, Dynamic Clipping, and LayerTuning-RL components. The results demonstrate that each component is individually effective. Specifically, introducing Dynamic-TreeRPO on top of the baseline effectively reduces training cost while maintaining performance across multiple evaluation metrics. Incorporating LayerTuning-RL and Dynamic Clipping addresses issues of inefficient exploration and training instability, enabling faster reward growth and smoother convergence.

To investigate the effect of the noise growth parameter $\beta$ in Equation 6 on the performance of our Dynamic-TreeRPO, we sampled a range of values evenly spaced between $0.3$ and $0.9$. The final results corresponding to these values of $\beta$ are summarized in Table 3. We observed that all three metrics initially exhibit a positive correlation with $\beta$, peaking at $\beta = 0.7$ before declining.

Table 2: Ablation experiments of Dynamic-TreeRPO

| Component | HPS-v2.1↑ | Pick Score↑ | ImageReward↑ |
|---|---|---|---|
| baseline | 0.313 | 0.227 | 1.088 |
| +Dynamic-Tree | 0.361 | 0.238 | 1.591 |
| +Dynamic-Clipping | 0.369 | 0.242 | 1.674 |
| +LayerTuning-RL | 0.385 | 0.251 | 1.770 |

Table 3: Comparison for noise growth hyperparameter $\beta$

| $\beta$ | HPS-v2.1↑ | Pick Score↑ | ImageReward↑ |
|---|---|---|---|
| 0.3 | 0.348 | 0.221 | 1.337 |
| 0.5 | 0.351 | 0.228 | 1.448 |
| 0.7 | 0.361 | 0.238 | 1.591 |
| 0.9 | 0.355 | 0.229 | 1.611 |

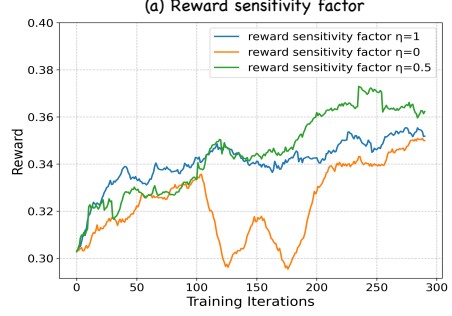
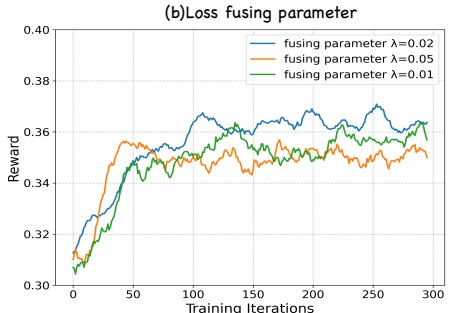

Figure 4: Ablation Studies on reward sensitivity factor and balancing parameter in LayerTuning-RL.

Furthermore, we conduct experiments to investigate the impact of the reward sensitivity factor $\eta$ in Equation 11 and the balancing parameter $\lambda$ in Equation 13. As shown in Figure 4(a), under the baseline + Dynamic Clipping framework, the setting $\eta=0$, which corresponds to the maximum clipping of the probability ratio, leads to noticeable instability during training, as reflected by the oscillatory learning curve. In contrast, the setting $\eta=0.5$ yields stable training dynamics while preserving sufficient exploration, indicating that the majority of generated samples effectively contribute to model updates. Figure 4(b) validates the effect of the balancing parameter $\lambda$. A larger $\lambda$ allows faster convergence in the early training stage, but imposes limitations in the later phase. Therefore, we set $\lambda=0.02$, which improves convergence without incurring additional computational cost and ensures the performance of RL through PRM.

## 5 CONCLUSION

In this paper, we introduce Dynamic-TreeRPO, a RL framework built upon a sliding tree structure. By incorporating dynamically adaptive clipping boundaries constrained by reward signals and integrating a training strategy combined with SFT, our approach accelerates training convergence while enabling efficient exploration of the search space. It effectively mitigates common issues in GRPO, such as catastrophic forgetting and inefficient exploration. Extensive experimental results demonstrate that our method achieves superior semantic consistency and visual quality in image generation.

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
