# A  APPENDIX

## A.1  LLM USAGE

We employed Large Language Models (LLMs) to assist in writing and polishing our manuscript. Specifically, we utilized them to correct grammatical errors, strengthen the logical flow and presentation of ideas, and enhance the overall readability of our manuscript, including this appendix. In particular, the use of the LLM was exclusively for linguistic refinement of the manuscript; it played no role in the ideation, methodology, or scientific analysis of the research.

# B  EXPERIMENT SETUP DETAILS

This section provides a detailed description of our experimental setup, including the model backbone used and the precise definitions of the evaluation metrics.

**Configuration of the reward function.** During training, we use HPS-v2.1, PickScore, and ImageReward as our reward functions, with a weighting ratio of 1:1:1.

**Model Backbone.** We employ FLUX.1-Dev as the backbone text-to-image generation model. This is a multimodal image generation and editing model with 12 billion parameters, supporting joint text-and-image inputs. It enables high-fidelity character consistency, localized editing, and style preservation. All experiments are initialized from the same pretrained checkpoint.

**Evaluation Metrics.** Regarding computational overhead, we adopt the number of function evaluations (NFE), which is further decomposed into $\text{NFE}_{\pi_{\theta_{\text{old}}}}$ and $\text{NFE}_{\pi_\theta}$. $\text{NFE}_{\pi_{\theta_{\text{old}}}}$ represents the number of forward propagation of the reference model for computing the policy ratio and generating images. $\text{NFE}_{\pi_\theta}$ is the number of forward propagation of the policy model solely for the policy ratio.

# C  MORE VISUALIZED RESULTS

💡 Prompt: Side-view blue-ice sneaker inspired by Spiderman created by Weta FX.

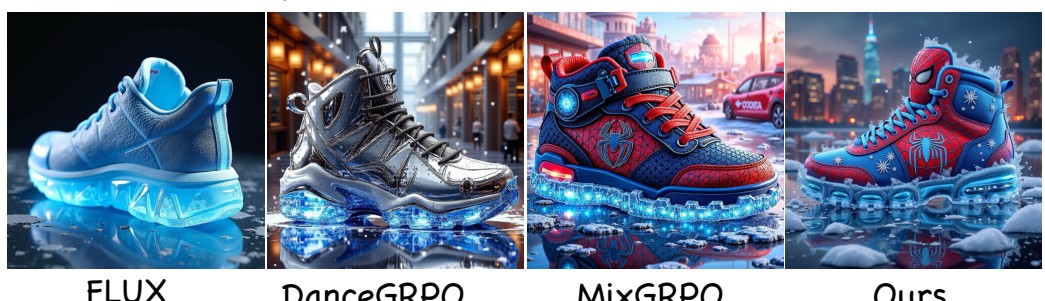

FLUX          DanceGRPO          MixGRPO          Ours

💡 Prompt: A small green dinosaur toy with orange spots standing on its hind legsand roaring with its mouth open.

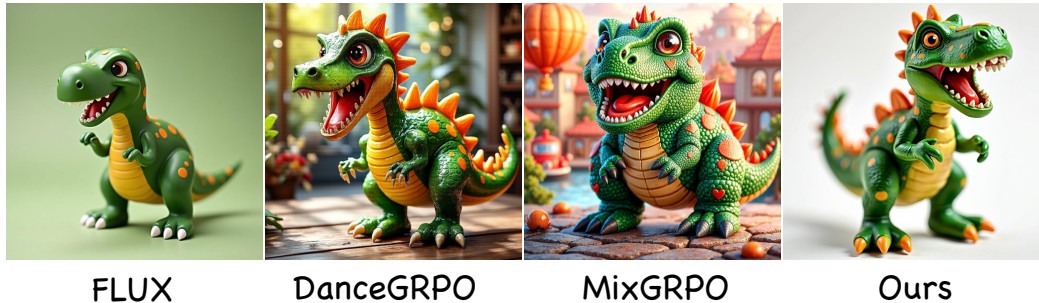

FLUX          DanceGRPO          MixGRPO          Ours

💡 Prompt: Cartoonish illustration of a sci-fi machine shop inside a shipping container.

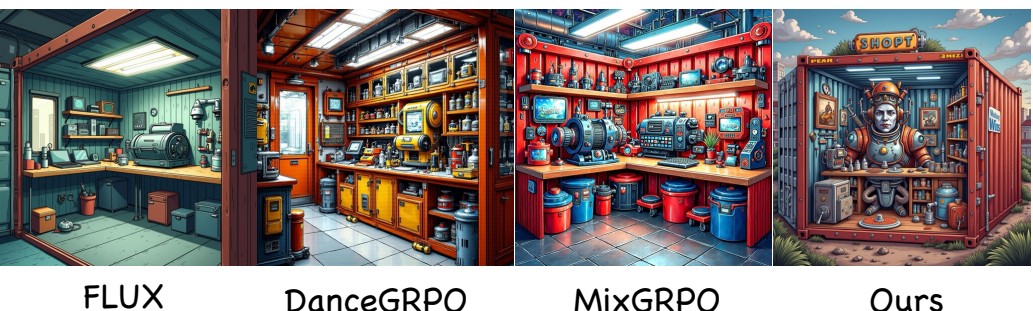

FLUX          DanceGRPO          MixGRPO          Ours

Figure 1: Comparison of the visualization results of FLUX, DanceGRPO, MixGRPO and Dynamic-TreeRPO.

💡 Prompt: A girl a white bear a camera a tea pot and a cup.

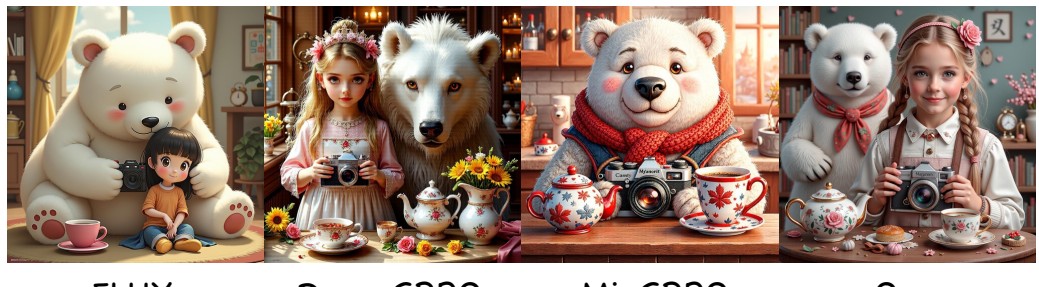

FLUX          DanceGRPO          MixGRPO          Ours

💡 Prompt: A great room with the living area in the foreground, dining table behind it and kitchen in the very back.

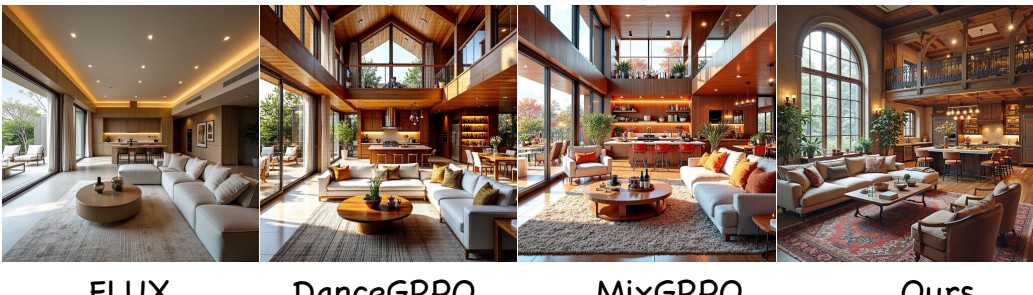

FLUX          DanceGRPO          MixGRPO          Ours

💡 Prompt: A little kid riding a skateboard down a sidewalk.

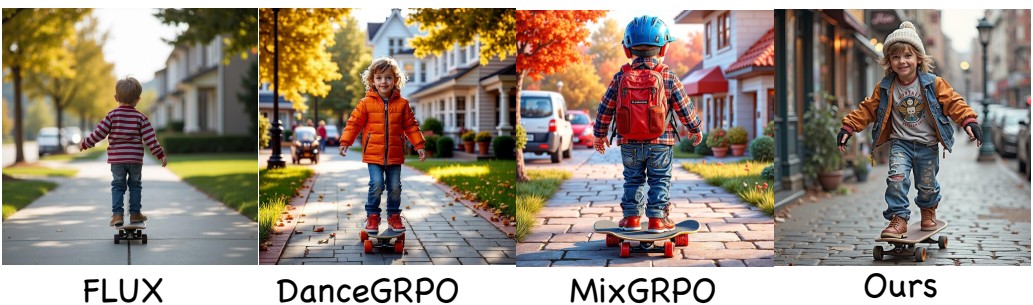

FLUX          DanceGRPO          MixGRPO          Ours

Figure 2: Comparison of the visualization results of FLUX, DanceGRPO, MixGRPO and Dynamic-TreeRPO.

Prompt: A lot of building on each side of the road, with a very curvy road in the middle.

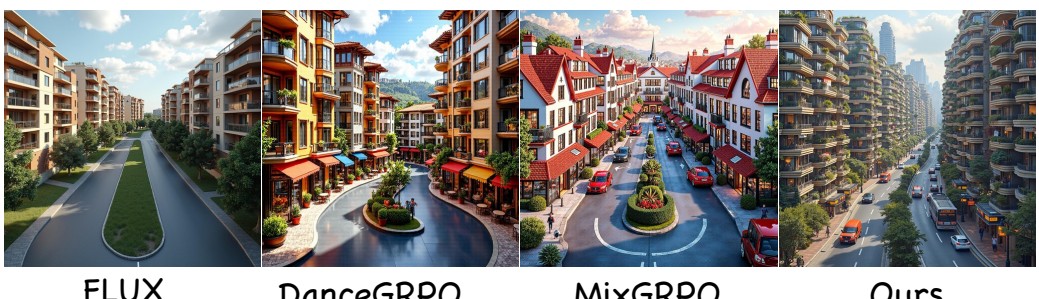

FLUX            DanceGRPO            MixGRPO            Ours

Prompt: A man placing a turkey in an oven with oven mitts.

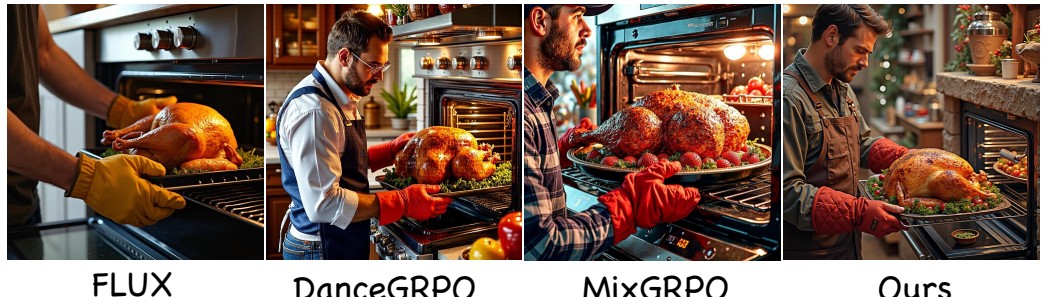

FLUX            DanceGRPO            MixGRPO            Ours

Prompt: A massive and brightly colored spacecraft in a deserted landscape, depicted in retro 1960s sci-fi art.

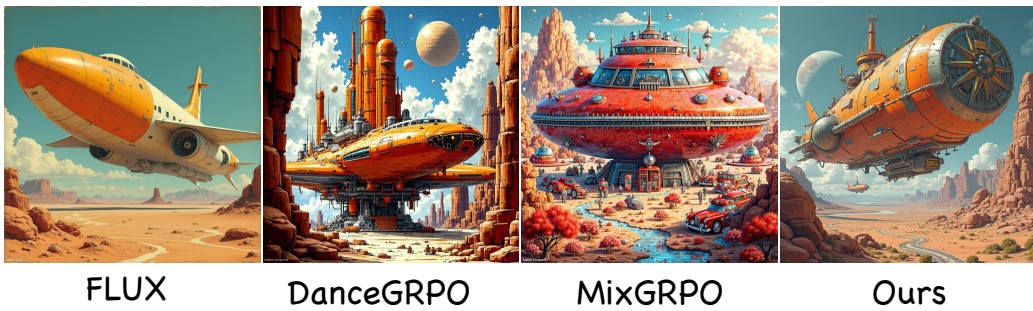

FLUX            DanceGRPO            MixGRPO            Ours

Figure 3: Comparison of the visualization results of FLUX, DanceGRPO, MixGRPO and Dynamic-TreeRPO.

💡 Prompt: A brown and black dog sticking its head out a window.

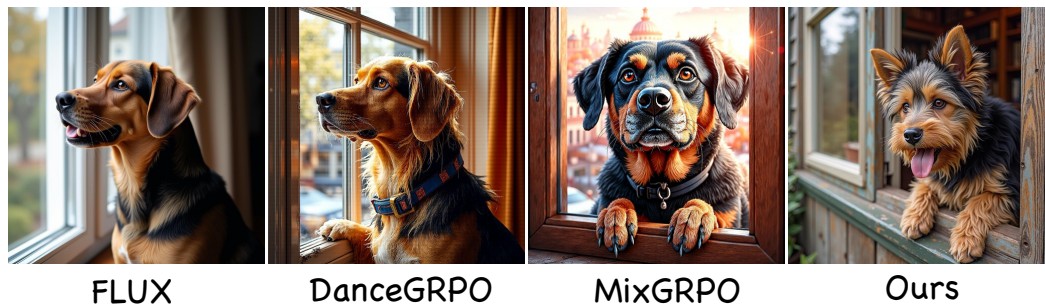

FLUX       DanceGRPO       MixGRPO       Ours

💡 Prompt: A castle is in the middle of a eurpean city.

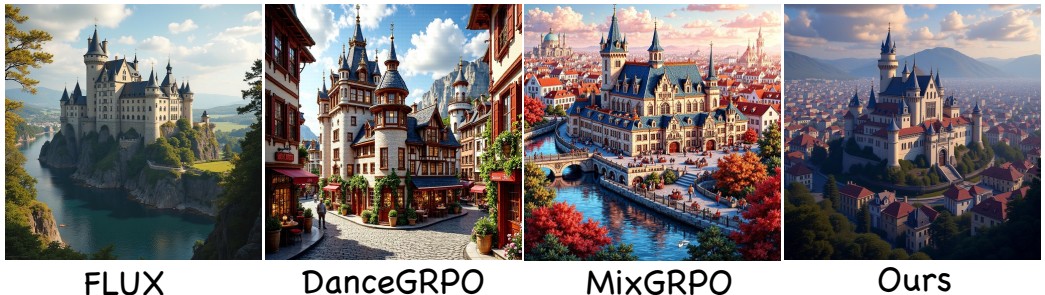

FLUX       DanceGRPO       MixGRPO       Ours

💡 Prompt: A fruit basket on a kitchen table with a Studio Ghibli reference.

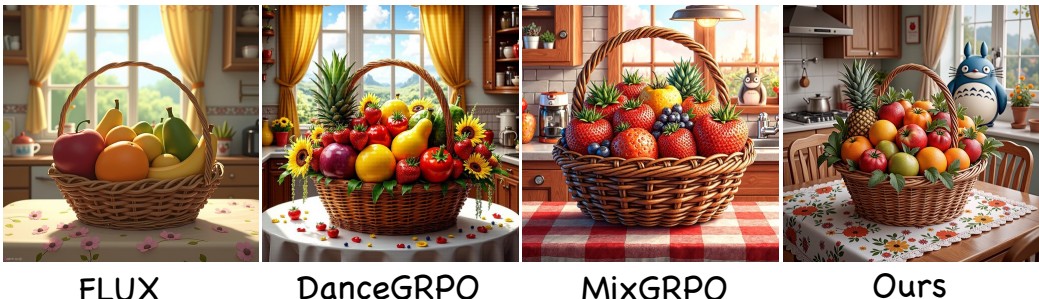

FLUX       DanceGRPO       MixGRPO       Ours

Figure 4: Comparison of the visualization results of FLUX, DanceGRPO, MixGRPO and Dynamic-TreeRPO.