# OpenReview forum: "Dynamic-TreeRPO: Breaking the Independent Trajectory Bottleneck with Structured Sampling"
_ICLR.cc/2026/Conference — Submitted to ICLR 2026_

### Official Review · Reviewer_Cwsi · 2025-10-25

**Soundness:** 2
**Presentation:** 3
**Contribution:** 2
**Rating:** 2
**Confidence:** 4

**Summary:**

This paper proposes Dynamic-TreeGRPO, a tree-structured search with dynamic noise, to address computational inefficiency and to separate training paradigms when applying RLHF to flow-based generative models. The proposed method gradually expands the search tree using a common trunk to reduce computational costs. Experiments demonstrate efficiency and state-of-the-art performance on benchmarks for human preference alignment, semantic consistency, and visual quality.

**Strengths:**

- The application of a tree-search-based approach in the image generation domain, to the best of my knowledge, is novel. Although similar approaches exist in other domains (see weaknesses).
- The proposed new training paradigm can provide a unified approach for SFT and RL.
- The computational cost compared to some baselines was demonstrated to be improved.

**Weaknesses:**

- The claims on improving efficiency are not well-motivated in the paper, nor are they fairly and comprehensively benchmarked against existing baselines. The paper claims a 50% efficiency improvement. Table 1 shows its iteration time (151s) is a ~37% reduction from MixGRPO (240s) and a ~53% reduction from DanceGRPO (323s). However, it is only a ~9% reduction from MixGRPO-Flash (166s). The source of the "50%" claim could be clearer, and the practical speed-up over other optimized baselines seems more modest. Indeed, as a tree-search-based approach, the number of nodes grows exponentially with tree depth, leading to significantly more training/finetuning time. It is indeed that, compared with DanceGRPO, the efficiency might be improved as only half of the nodes are needed, but compared with vanilla GRPO, e.g., [1], the number of forward passes in a single data-point optimization is still significantly larger due to the need to expand the tree. As [1] was never compared as a baseline, it remains unclear to me whether the proposed approach is truly efficient.

- Tree-search-based RL approaches for generative models are not uncommon. For example, [2] combined tree-search with diffusion for planning in RL tasks. Fundamentally, the proposed TreeGRPO also follows the same spirit.

- The LayerTuning-RL component uses the best sample from the current group ($V_t^*$) as the "pseudo target" for its SFT loss. What happens if the entire group of samples is poor or low-quality? The model would essentially be learning to imitate the "best of the worst," which could lead to a local optimum or mode collapse rather than true quality improvement.

- The authors use a binary tree of a fixed depth ($d=4$). How was this structure chosen? How does the performance change with deeper or wider (e.g., ternary) trees? A fixed, shallow tree might limit the complexity of the exploration, and the optimal depth is likely task-dependent.

- The paper proposes a linear increase in noise intensity based on tree depth (Equation 6). This is an interesting heuristic, but is it the most effective? Were other functions (e.g., exponential, logarithmic) explored to see if they could provide a better exploration/exploitation trade-off?

- The number of RL baselines compared in the paper is too limited to demonstrate the true effectiveness of the proposed approach compared to existing baselines. For example, [1] is a closely related baseline that directly applies GRPO for flow-based generative models. In addition, as various DPO (direct preference optimization) approaches already exist and achieve decent results, at least one of them should be compared.

[1] Liu, Jie, et al. "Flow-grpo: Training flow matching models via online rl." arXiv preprint arXiv:2505.05470 (2025).

[2] Yoon, Jaesik, et al. "Monte carlo tree diffusion for system 2 planning." arXiv preprint arXiv:2502.07202 (2025).

**Questions:**

Please refer to the weaknesses for more details. In addition:

- Can the authors discuss the connection of the proposed method to existing tree-search-based methods like [2]?
- Can you give an intuitive interpretation of why LayerTuning-RL can lead to better performance?

---

> ### Author Response · Authors · 2025-11-27
> **Author Response**
>
> Thank you for your valuable feedback we’ve addressed your suggestions below.
>
> **Response:**
>
> **1. About efficiency improvement.**
>
> The efficiency advantage of our method arises from the following: once the tree depth is fixed (and consequently the number of groups associated with GRPO is determined), we leverage shared prefixes to reduce the number of evaluations of  $\text{NFE}\_{\pi\_{\theta\_{\text{old}}}}$   from $2^{d-1} \* T$  to $τ+ 2^{d-1}\*(T-τ)$.  The root node slides from 0 to $T-d$ with an average of $\frac{(2^{d-1} - 1)(T - 4)}{2}$ slides during training. Based on this reduction, we made the claim of nearly 50% improvement in training efficiency.
>
> However, in Table 1, MixGRPO-Flash uses T=16, our method still uses T=25, this discrepancy makes the comparison unfair.
>
> **2. Relationship with MCTS[1]**
>
> The [1] approach reformulates the denoising process as a tree-based rollout procedure with preference-guided tree expansion to identify the optimal path. In contrast, our method leverages a tree structure—combined with shared prefixes and dynamic noise—to accelerate GRPO training and enlarge the exploration space. In short, MCTS uses trees to better utilize the model, whereas Dynamic-TreeRPO uses trees to better train the model.
>
> **3. About the LayerTuning-RL**
>
> Your question cuts straight to the heart of our core idea—the fundamental motivation behind LayerTuning-RL: to transform SFT from an isolated pre-training phase into a dynamic “knowledge anchor” that persists throughout the entire RL  training process. In simple terms, during each RL exploration step, we apply preference learning to the best-performing sample observed in that iteration.
>
> Regarding your concern about “learning the best of the worst, leading to local optima or model collapse”  it’s important to note that, at its core, RL merely seeks to maximize the likelihood of currently observed good samples. Moreover, in LayerTuning-RL, the SFT loss serves only as an auxiliary term—it does not dominate the learning process.It provides a gentle “pull” that helps keep the RL policy from deviating too far from high-quality trajectories.
>
> Furthermore, as training progresses and overall rewards improve, the probability of generating high-quality samples naturally increases. Additionally, our dynamic clipping boundaries ensure that even in low-reward regions, the model can still perform bold exploration, effectively preventing premature convergence to suboptimal policies.
>
> **4. About the hyperparameters related of the  tree structure**
>
> Tree depth essentially represents the number of exploration groups in GRPO. While deeper trees can indeed provide a larger exploration space, they also lead to a sharp increase in computational cost. Based on extensive experimentation, we adopted a depth of d=4 , which strikes a balance between performance and efficiency. Multi-branch trees are certainly a promising direction, and we plan to investigate the impact of different branching factors in future work.
>
> **5. About the design of noise intensity**
>
> Thank you for pointing this out. We chose a linearly increasing schedule primarily as a simple and interpretable heuristic. We did experiment with other forms, such as exponential schedules, but found that introducing excessively large differences in noise levels across layers often led to severe artifacts and distorted generation results. We will explore alternative approaches in future work.
>
> **6. About the comparison with FlowGRPO and DPO**
>
> Methodologically, DanceGRPO and FlowGRPO are largely analogous. Both methods convert the sampling process of a flow-matching model into an equivalent full-step SDE sampling, thereby introducing the necessary stochasticity for reinforcement learning exploration. While they differ slightly in their mathematical formulation, their primary distinction lies in the implementation details: DanceGRPO performs full fine-tuning on the Flux model, whereas FlowGRPO utilizes LoRA for fine-tuning on SD3.5-M.
>
> To ensure a fair comparison, we adopted the official configurations for each method. We have added the test results of Flow-GRPO trained on the HPDv21 dataset, achieving scores of 0.364 on HPS-v2.1, 0.238 on Pick Score, and 1.558 on ImageReward, respectively.
>
> About DPO-based methods, we agree that including them in the comparison would be valuable. However, to the best of our knowledge, there is currently no widely accepted DPO baseline that directly competes with GRPO methods on standard text-to-image (T2I) benchmarks.  Should such a baseline become available or be brought to our attention, we would be happy to incorporate it into our comparisons. Furthermore, such as Flow-GRPO, DanceGRPO, TempFlow-GRPO, and MixGRPO also do not include comparisons with DPO-based approaches.
>
> **I look forward to your response. Your timely response would be greatly appreciated !**
>
> [1] Yoon, Jaesik, et al. "Monte carlo tree diffusion for system 2 planning." arXiv preprint arXiv:2502.07202.

---

### Official Review · Reviewer_LEpw · 2025-10-27

**Soundness:** 2
**Presentation:** 2
**Contribution:** 2
**Rating:** 2
**Confidence:** 4

**Summary:**

The paper proposes Dynamic-TreeRPO, an RL training framework for flow-matching T2I models that (i) replaces independent trajectory sampling with a tree-structured sampler using a sliding window where SDE noise is injected, sharing prefix computations for efficiency; (ii) introduces dynamic, reward-conditioned clipping for GRPO; and (iii) integrates SFT as a LayerTuning-RL auxiliary loss, treating SFT as a progress-reward model (PRM) that supervises nodes along the tree. Claimed benefits include ~50% faster training iterations and higher scores on HPS-v2.1, PickScore, and ImageReward, with experiments primarily on FLUX.1-dev and HPDv2.1 prompts.

**Strengths:**

Clear engineering idea: the tree sampler amortizes compute by sharing prefixes; ablations isolate contributions (Dynamic-Tree, Dynamic-Clipping, LayerTuning-RL).
Reported efficiency gains with stable reward curves and consistent improvements on automatic preference metrics.

**Weaknesses:**

Limited conceptual novelty vs. recent GRPO variants.
The tree + sliding-window SDE is close in spirit to MixGRPO’s sliding-window stochasticity and TempFlow-GRPO’s trajectory branching/credit assignment; Dynamic-TreeRPO reads as an engineering re-packaging rather than a new optimization principle. A convincing distinction (either theoretically or via stronger baselines) is missing.

Optimization is confined to SDE steps in the window.
By design, RL updates only occur inside the tree interval; outside it, ODE sampling is used and not optimized. This structurally restricts policy learning to a subset of timesteps and may mis-assign credit across the generative trajectory. The objective in Eq. (9) explicitly sums GRPO terms only over $t\inS_{tree}$. Strong evidence is needed that this does not harm global alignment or late-stage refinements.

Ambiguous/dated reward-modeling and missing reward baselines.
The paper reports evaluation with HPS-v2.1, PickScore, and ImageReward, but does not clearly state which reward model(s) drive training nor why stronger contemporary RMs are excluded. Modern alternatives such as VisionReward (fine-grained, multi-dimensional), or VLM-based/zero-shot RMs are not used or compared, weakening claims about alignment under stronger judges.

SFT/PRM supervision is under-specified and risks label leakage.
LayerTuning-RL defines the SFT target at each step as the max-reward path’s prediction $V^*_t$, i.e., model-generated pseudo-labels, not human-labeled images. The paper does not clarify where labeled images come from (if any), how pseudo-label bias is controlled, or how this differs from reward-weighted regression. The claim that this mitigates reward hacking without KL is unsubstantiated beyond small-scale metrics.

Missing or incomplete baseline comparisons.
Beyond Flux, DanceGRPO, and MixGRPO, several relevant baselines are absent:
Flow-GRPO (first online RL for flow matching) and TempFlow-GRPO (temporally aware branching).
Diffusion-DPO (non-RL preference optimization) and DPOK/DPoK-style methods for T2I alignment.
DDPO / RL for diffusion with explicit reward design, which is a strong alternative alignment route.
Without these, it’s hard to attribute the reported gains to the tree structure rather than to known improvements from recent GRPO/DPO lines.

Evaluation scale and generality are limited.
Training uses a subset of HPDv2.1 prompts and 400 test prompts; no human A/B tests, no safety/failure analyses, and no robustness to adversarial prompts or typography tasks. This makes the “SoTA” claim fragile.

Dynamic clipping lacks principled comparison to known variants.
The paper positions reward-conditioned clipping as stabilizing, yet omits direct comparisons with DAPO/DCPO-style adaptive clipping (strong baselines in RLHF), even as related work acknowledges them conceptually. A head-to-head would clarify whether reward-conditioning brings new benefits beyond token-probability-aware schemes.

**Questions:**

Exactly which RM(s) produce the training rewards (not just evaluators)? Why not include VisionReward or VLM-based RMs, and what happens if you swap the judge (HPS→VisionReward, etc.)? Please add ablations with multiple modern RMs and report reward-hacking diagnostics (diversity, FID, failure rate).

SFT supervision source: LayerTuning-RL uses $V^*_t$ from the best in-group sample as the SFT target. Does any external labeled (image,label) data play a role? If not, how do you prevent confirmation bias from pseudo-labels? Please clarify data provenance, label construction, and compare against standard reward-weighted likelihood / ReFL.

Why is optimization restricted to $S_{tree}$? Provide experiments where you (a) shift the window across timesteps, (b) expand to all steps, and (c) vary the branching schedule (vs. TempFlow-GRPO). Report effects on long-horizon credit assignment and text rendering tasks.

Please add Flow-GRPO, TempFlow-GRPO, Diffusion-DPO, and DPOK as baselines under matched compute/hardware, plus ablations against adaptive-clipping methods (e.g., DCPO/DAPO families). Without these, it’s unclear that Dynamic-TreeRPO itself—not broader training choices—drives the gains.

---

> ### Author Response · Authors · 2025-11-27
> **Author Response**
>
> We sincerely thank you for taking the time to provide detailed and constructive feedback on our submission. Your comments are highly valuable and have significantly helped us identify areas for improvement in our work. Below, we address your suggestions and concerns.
>
> **Response:**
>
> **1. Limited conceptual novelty vs. recent GRPO variants**
>
> Our work shares a common goal with MixGRPO, DanceGRPO, and TempFlow-GRPO: achieving an effective trade-off between training efficiency and exploration space through thoughtful design. Compared to other methods, our key innovation lies in using a sliding tree structure to improve training efficiency, and addressing the limited exploration diversity inherent in tree-based structures by introducing dynamic noise and a PRM-based strategy—features not well captured in prior work. More importantly, effectively integrating these components so they complement and reinforce each other requires deep research insight, not merely straightforward engineering repackaging.
>
> **2. Optimization is confined to SDE steps in the window**
>
> Applying SDE on a sliding window is a common approach to improve exploration efficiency, as demonstrated in prior works such as Flow-GRPO and MixGRPO. Building upon the sliding window, we introduce a tree-structured design along with a tailored noise augmentation strategy, enabling more fine-grained and effective spatial exploration while maintaining high exploration efficiency.
>
> **3. Ambiguous/dated reward-modeling**
>
> In our current experimental setup, we strictly follow the standard configurations of each benchmark and use the same training and evaluation reward functions as the baselines. We will clearly specify these experimental details in the paper. The VisionReward and VLM-based or zero-shot reward models (RMs) you mentioned have not been used in recent works, and introducing additional RMs could compromise the fairness of the comparison.
>
> **4. SFT/PRM supervision is under-specified and risks label leakage**
>
> In LayerTuning-RL, we use no external data. Instead, we select the sample with the highest final reward from the N trajectories generated by ${\pi_{\theta_{\text{old}}}}​​$ in each iteration. Critically, no human-annotated data is involved. The core idea of LayerTuning-RL is to transform  SFT from a standalone pre-training phase into a dynamic “knowledge anchor” throughout the entire RL training process. Specifically, during each RL exploration step, we apply preference toward the best-performing sample from that iteration. Notably, the SFT loss in LayerTuning-RL serves only as an auxiliary term—it does not dominate learning but provides a gentle pull to prevent the RL policy from deviating from high-quality trajectories. As training progresses and overall rewards improve, the probability of generating high-reward samples also increases.
>
> **5. Missing or incomplete baseline comparisons**
>
> Methodologically, DanceGRPO and FlowGRPO are largely analogous. Both methods convert the sampling process of a flow-matching model into an equivalent full-step SDE sampling, thereby introducing the necessary stochasticity for reinforcement learning exploration. While they differ slightly in their mathematical formulation, their primary distinction lies in the implementation details: DanceGRPO performs full fine-tuning on the Flux model, whereas FlowGRPO utilizes LoRA for fine-tuning on SD3.5-M.
>
> To ensure a fair comparison, we adopted the official configurations for each method. We have added the test results of Flow-GRPO trained on the HPDv21 dataset, achieving scores of 0.364 on HPS-v2.1, 0.238 on Pick Score, and 1.558 on ImageReward, respectively.
>
> About DPO-based methods, we agree that including them in the comparison would be valuable. However, our current focus is primarily on comparing against GRPO-based approaches. Moreover, to the best of our knowledge, there is currently no widely accepted DPO baseline that directly competes with GRPO methods on standard text-to-image (T2I) benchmarks. Should such a baseline become available or be brought to our attention, we would be happy to incorporate it into our comparisons.You noted that methods such as Flow-GRPO, DanceGRPO, TempFlow-GRPO, and MixGRPO also do not include comparisons with DPO-based approaches.
>
> **6. Dynamic clipping lacks principled comparison & Evaluation scale and generality are limited**
>
> We agree with this point. DAPO and DCPO are important works in the RLHF literature, and we will add a comparison against these methods. We are also conducting human evaluations, and the paper will be updated once these studies are complete.
>
> **I look forward to your response. Your timely response would be greatly appreciated !**

---

### Official Review · Reviewer_nhZs · 2025-10-31

**Soundness:** 3
**Presentation:** 3
**Contribution:** 3
**Rating:** 6
**Confidence:** 3

**Summary:**

This paper introduces Dynamic-TreeRPO, a reinforcement learning (RL) framework designed for flow-matching text-to-image models. The method utilizes a sliding-window tree-structured SDE process with dynamic noise intensities, thereby reducing the computational overhead. Furthermore, the authors present a novel training paradigm named LayerTuning-RL, which unifies supervised fine-tuning (SFT) and reinforcement learning into a single optimization objective. Experiments demonstrate that the proposed method improves training efficiency and achieves superior performance compared to previous methods (DanceGRPO and MixGRPO).

**Strengths:**

1. Improving the efficiency of GRPO-based RL framework in flow matching text-to-image models is an important research topic. The proposed tree-structured search method provides a reasonable and well-motivated solution.
2. The idea of LayerTuning-RL, i.e., using the trajectory that obtains the highest reward within each group as the supervising target, is conceptually appealing and offers an interesting perspective on integrating SFT and RL.

**Weaknesses:**

The proposed method introduces a number of hyperparameters, such as the depth of trajectory trees $d$, noise growth magnitude $\beta$ in Equation (6), reward sensitivity factor $\eta$ in Equation (11), clipping threshoulds $\varepsilon_\text{low}$ and $\varepsilon_\text{high}$ in Equation (11), and loss coefficient $\lambda$ in Equation (13). These hyperparameters may require careful tuning, imposing additional burdens in practice.

**Questions:**

1. How sensitive is the proposed method to the hyperparameters mentioned in the “Weaknesses” section?
2. How is the tree depth $d$ determined? Is it feasible to use a wider tree structure instead of a binary tree?
3. What motivates the design of the function $g_t(k)$ in Equation (6)? More specifically, why is it designed to linearly increase with the normalized depth? Have you tried other formulations such as those decrease with depth?
4. Can the technique of using different noise intensities be applied to MixGRPO and DanceGRPO, to enhance their diversity and performance?

---

> ### Author Response · Authors · 2025-11-27
> **Author Response**
>
> We sincerely thank you for taking the time to provide detailed and constructive feedback on our submission. Your comments are highly valuable and have significantly helped us identify areas for improvement in our work. Below, we address your suggestions and concerns.
>
> **Response:**
>
> **1. About the design of hyperparameters**
>
> The parameters $ε\_{low}$,$ε\_{high}$ and $η$ are standard in vanilla GRPO and its extensions, and we adopt their values based on established tuning practices in the field.
>
> In our method, we carefully study the hyperparameters depth $d$、$β$ and $λ$. Experiments show that, besides $d$ and $β$ which significantly affect both the speed and quality of policy learning—the other parameters perform well with moderate default values.
>
> **2. About binary tree**
>
> As you rightly pointed out，binary trees are a special case of n-ary trees. While n-ary trees can achieve the same number of exploration paths with fewer depth levels, their exploration diversity decreases. We adopt binary trees as a trade-off between computational cost and diversity. In future work, we will investigate how to design n-ary tree-based exploration strategies that offer higher diversity.
>
> **3. About  Equation (6)**
>
> Our motivation is to maximize diversity among sibling nodes at the same level of the binary tree. Since siblings share a common ancestor, their diversity is inherently limited. To address this, we introduce artificial perturbations that increase randomness with tree depth. Using a decreasing perturbation schedule with depth would reduce randomness, resulting in more similar sampled nodes.
>
> **4. Application of noise intensity**
>
> We also tested different noise intensities in Mix-GPPO. The differences are clearly visible in the std values shown in Figure 1, and qualitative analysis further confirms the improvement in diversity.
>
> **I look forward to your response. Your timely response would be greatly appreciated !**

---

### Official Review · Reviewer_Rrvw · 2025-11-01

**Soundness:** 2
**Presentation:** 2
**Contribution:** 3
**Rating:** 6
**Confidence:** 4

**Summary:**

Dynamic-TreeRPO is a tree-structured reinforcement learning framework for flow matching models that uses a binary tree and LayerTuning-RL to jointly enhance sampling efficiency, exploration diversity, and achieve state-of-the-art performance across multiple benchmarks.

**Strengths:**

Improving sampling efficiency is crucial in reinforcement learning tasks. The proposed binary-tree-based sampling strategy effectively accelerates sampling while maintaining diversity, representing a practical and well-motivated contribution.

**Weaknesses:**

The proposed algorithm consists of three components, with the binary tree sampling serving as its core idea. However, from Table 2, the performance gains from the other two improvements are also considerable. Combined with the results in Table 1, the performance of the binary tree and MixGRPO variants appears comparable. It is unclear whether the iteration time reported in Table 1 accounts only for the binary tree module or also includes the SL component, which would also increase training time.

**Questions:**

What is the starting step of the denoising time range parameter?

If only the binary tree sampling is used, would the in-group diversity decrease compared to the MixGRPO formulation? It might be valuable to quantify this with appropriate diversity metrics to support the claim.

---

> ### Author Response · Authors · 2025-11-27
> **Author Response**
>
> We sincerely thank you for taking the time to provide detailed and constructive feedback on our submission. Your comments are highly valuable and have significantly helped us identify areas for improvement in our work. Below, we address your suggestions and concerns.
>
> **Response:**
>
> **1.About training time**
>
> The efficiency advantage of our method arises from the following: once the tree depth is fixed (and consequently the number of groups associated with GRPO is determined), we leverage shared prefixes to reduce the number of evaluations of $\text{NFE}\_{\pi\_{\theta\_{\text{old}}}}$ from $2^{d-1}\*T$ to $τ+ 2^{d-1}\*(T-τ)$ .  The root node slides from 0 to $T-d$ with an average of $\frac{(2^{d-1} - 1)(T - 4)}{2}$ slides during training.
>
> The other two improvements also show substantial gains over the baseline. For the SFT module, we efficiently reuse the per-timestep outputs from the forward inference phase, so it adds no overhead to inference time—only a small increase in loss computation time.
>
> **2. About  denoising time range parameter**
>
> In the paper, we use $τ ∈ [0, T − d]$ to denote the starting step of the denoising time range parameter, which is randomly selected in each iteration.
>
> **3. About  group diversity**
>
> Compared to MixGRPO, our method improve diversity. This is enabled by two key mechanisms: (1) a dynamic noise strategy that expands the exploration space by adaptively adjusting the noise at each level of the binary tree, and (2) a dynamic clipping mechanism in the reward function that adjusts exploration bounds in real time based on reward magnitude. See Sections 3.2 and 3.3 for details.
>
> Thanks to these mechanisms, our approach achieves higher diversity than MixGRPO and other baselines. Qualitative evidence is shown in Figure 1 via intra-group variance.
>
> **I look forward to your response. Your timely response would be greatly appreciated !**

---

### Meta-Review · Area_Chair_uocW · 2026-01-04

**Summary:**

This paper presents a new RL approach for flow matching models, named Dynamic-TreeRPO. Dynamic-TreeRPO uses a tree-structured search with dynamic noise intensities along depth. By sharing prefix paths of the tree, Dynamic-TreeRPO improves the efficiency of RL sampling. Furthermore, this paper integrate SFT with RL to improve the diversity of search directions. In the initial review, this paper is rated by 6,6,2 and 2. The concerns mainly focus on hyperparameter analysis, the comparison with previous works, and limited novelty. During rebuttal, the authors provides explanations to the questions. However, AC confirms that the explanations are quite weak without enough support, and the concerns are not fully addressed. Therefore, AC recommends this paper as **Reject**.

**Reviewer Concerns:**

In the initial review, this paper is rated by 6,6,2 and 2. The concerns mainly focus on hyperparameter analysis, the comparison with previous works, and limited novelty.

During rebuttal, the authors provide explanation on why they choose such hyperparameters, such as depth. However, the explanation is quite weak due to the lack of support.

For the comparison with previous works, including the reward baselines and other RL methods, the authors do not provide comprehensive comparison.

For the limited novelty, although the authors re-claim their novelty, the approach still shares similarities with previous works.

Therefore, AC confirms that all the main concerns are not addressed.

**Reviewer Scores:**

**Reviewer Rrvw** keeps 6 or degrades to 4. Review Rrvw does not have the chance to reply to the authors. The review of Reviewer Rrvw is kind of weak, and the reviewer's concerns are mainly addressed but the reviewer might change the mind to a negative score after reading other reviews.

**Reviewer nhZs** degrades to 4. Review nhZs does not have the chance to reply to the authors. AC confirms that the reviewer's concerns are not addressed.

**Reviewer LEpw** keeps 2. Review LEpw does not have the chance to reply to the authors. AC confirms that the reviewer's concerns are not addressed.

**Reviewer Cwsi** keeps 2. Review Cwsi does not have the chance to reply to the authors. AC confirms that the reviewer's concerns are not addressed.

---

### Decision · Program_Chairs · 2026-01-26

Reject